# A *DSG1* Frameshift Variant in a Rottweiler Dog with Footpad Hyperkeratosis

**DOI:** 10.3390/genes11040469

**Published:** 2020-04-24

**Authors:** Katherine A. Backel, Sarah Kiener, Vidhya Jagannathan, Margret L. Casal, Tosso Leeb, Elizabeth A. Mauldin

**Affiliations:** 1School of Veterinary Medicine, University of Pennsylvania, Philadelphia, PA 19104, USA; kbackel@metro-vet.com (K.A.B.); casalml@vet.upenn.edu (M.L.C.); emauldin@vet.upenn.edu (E.A.M.); 2Institute of Genetics, Vetsuisse Faculty, University of Bern, 3001 Bern, Switzerland; sarah.kiener@vetsuisse.unibe.ch (S.K.); vidhya.jagannathan@vetsuisse.unibe.ch (V.J.); 3Dermfocus, University of Bern, 3001 Bern, Switzerland

**Keywords:** *Canis lupus familiaris*, whole-genome sequence, animal model, genodermatosis, skin, dermatology, keratinocyte, SAM syndrome, precision medicine

## Abstract

A single male Rottweiler dog with severe footpad hyperkeratosis starting at an age of eight weeks was investigated. The hyperkeratosis was initially restricted to the footpads. The footpad lesions caused severe discomfort to the dog and had to be trimmed under anesthesia every 8–10 weeks. Histologically, the epidermis showed papillated villous projections of dense keratin in the stratum corneum. Starting at eight months of age, the patient additionally developed signs consistent with atopic dermatitis and recurrent bacterial skin and ear infections. Crusted hyperkeratotic plaques developed at sites of infection. We sequenced the genome of the affected dog and compared the data to 655 control genomes. A search for variants in 32 candidate genes associated with human palmoplantar keratoderma (PPK) revealed a single private protein-changing variant in the affected dog. This was located in the *DSG1* gene encoding desmoglein 1. Heterozygous monoallelic *DSG1* variants have been reported in human patients with striate palmoplantar keratoderma I (SPPK1), while biallelic *DSG1* loss of function variants in humans lead to a more pronounced condition termed severe dermatitis, multiple allergies, and metabolic wasting (SAM) syndrome. The identified canine variant, *DSG1*:c.2541_2545delGGGCT, leads to a frameshift and truncates about 20% of the coding sequence. The affected dog was homozygous for the mutant allele. The comparative data on desmoglein 1 function in humans suggest that the identified *DSG1* variant may have caused the footpad hyperkeratosis and predisposition for allergies and skin infections in the affected dog.

## 1. Introduction

The skin forms an essential barrier against the environment. In humans, the soles of the feet and the palms of the hands are covered by the specially structured palmoplantar epidermis, which has to bear the strongest mechanical forces of the entire skin. Genodermatoses characterized by altered structural and junctional proteins of these specialized regions comprise the palmoplantar keratodermas (PPK), a diverse group of inherited disorders collectively characterized by excessive or abnormal thickening of the palmoplantar skin. Variants in at least 32 genes have been shown to cause different forms of isolated or syndromic PPK in humans (Table 1) [1,2].

Footpad hyperkeratosis in dogs is a genetically heterogenous group of inherited diseases corresponding to human PPK. So far, causative genetic variants for two different forms of canine footpad hyperkeratosis have been reported. Hereditary footpad hyperkeratosis (HFH) in Irish Terriers and Kromfohrländer dogs is caused by a variant in the *FAM83G* gene [36] encoding a protein involved in BMP and WNT signaling [37,38,39]. The syndromic HFH phenotype is characterized by an orthokeratotic hyperplasia of the footpad epidermis in combination with an irregular hair morphology. *FAM83G* variants have also been described in human patients with PPK and exuberant scalp hair [12] and the wooly mouse mutant [40]. A *KRT16* frameshift variant has been reported in Dogues de Bordeaux with focal nonepidermolytic footpad hyperkatosis [41]. Interestingly, this disease is inherited as an autosomal recessive trait, while the human forms of *KRT16*-associated focal nonepidermolytic PPK typically are inherited as autosomal dominant traits [25,26].

This study was initiated after an owner reported a juvenile male Rottweiler dog suffering from footpad hyperkeratosis. The goal of the study was to characterize the clinical and histopathological phenotype and to identify a possible underlying causative genetic defect.

## 2. Materials and Methods

### 2.1. Ethics Statement

All animal experiments were performed according to local regulations. The dog in this study is privately owned and was examined with the consent of the owner. The "Cantonal Committee for Animal Experiments" approved the collection of blood samples (Canton of Bern; permit 75/16).

### 2.2. Animal Selection

A male Rottweiler dog with footpad hyperkeratosis was investigated. Footpad biopsies were collected at initial presentation to rule out infectious and inflammatory causes of hyperkeratosis. The clinical presentation was inconsistent with other causes of secondary hyperkeratosis. An EDTA blood sample was collected for genomic DNA isolation. Additionally, we used 15 blood samples from other Rottweilers, which had been donated to the Vetsuisse Biobank. They represented population controls without reports of footpad hyperkeratosis. The photo of the control Rottweiler, shown in Figure 1B, represents a stock photo from the University of Pennsylvania (UPENN) veterinary hospital. This dog was not genotyped. The biopsy of the control dog, shown in Figure 2A, originates from a six-month-old healthy Beagle that was part of another IACUC-approved study at the UPENN School of Veterinary Medicine.

### 2.3. Histopathological Examinations

Two 6 mm punch biopsies from the footpads were obtained under general anesthesia. The samples were fixed in 10% neutral buffered formalin and routinely processed, including staining with hematoxylin and eosin.

### 2.4. DNA Extraction

Genomic DNA was isolated from EDTA blood with the Maxwell RSC Whole Blood Kit using a Maxwell RSC instrument (Promega, Dübendorf, Switzerland).

### 2.5. Whole-Genome Sequencing

An Illumina TruSeq PCR-free DNA library with ~500 bp insert size of the affected dog (RO015) was prepared. We collected 329 million 2 × 150 bp paired-end reads on a NovaSeq 6000 instrument (37x coverage). Mapping and alignment were performed as described [42]. The sequence data were deposited under the study accession PRJEB16012 and the sample accession SAMEA6249501 at the European Nucleotide Archive.

### 2.6. Variant Calling

Variant calling was performed using GATK HaplotypeCaller [43] in gVCF mode as described [42]. To predict the functional effects of the called variants, SnpEff [44] software together with NCBI annotation release 105 for the CanFam3.1 genome reference assembly was used. For variant filtering we used 655 control genomes (Appendix A).

### 2.7. Gene Analysis

We used the CanFam3.1 dog reference genome assembly and NCBI annotation release 105. Numbering within the canine *DSG1* gene corresponds to the NCBI RefSeq accession numbers NM_001002939.1 (mRNA) and NP_001002939.1 (protein).

### 2.8. Sanger Sequencing

The *DSG1*:c.2541_2545delGGGCT variant was genotyped by direct Sanger sequencing of PCR amplicons. A 402 bp (or 397 bp in case of the mutant allele) PCR product was amplified from genomic DNA using AmpliTaqGold360Mastermix (Thermo Fisher Scientific, Waltham, MA, USA) together with primers 5‘-GAG CAC TGA ACC GAT TTG CC -3‘ (Primer F) and 5’- GGC ATA GTC AAA GAG GTG GGT-3’ (Primer R). After treatment with exonuclease I and alkaline phosphatase, amplicons were sequenced on an ABI 3730 DNA Analyzer (Thermo Fisher Scientific). Sanger sequences were analyzed using the Sequencher 5.1 software (GeneCodes, Ann Arbor, MI, USA).

## 3. Results

### 3.1. Clinical Examination

A six-month-old male intact Rottweiler dog presented for evaluation of thick, rapidly growing footpads and discomfort (shifting stance, unwilling to stand for long periods). An unusual appearance of the pads (described as “dryness”) had been first noted by the owners when the dog was obtained at eight weeks of age. At the time of the initial presentation, the patient was otherwise healthy with no other systemic or dermatologic signs. On examination, all pads on all feet were markedly thickened by dense mounds of adherent keratin (Figure 1). The digital pads and metatarsal/metacarpal pads were the most severely affected. Fissures and mobile keratin ridges were present along the edges of the pads. There was mild diffuse scale over the trunk, which was considered to be within normal limits.

Regular application of moisturizers (urea), keratolytic (propylene glycol), and keratoplastic agents (salicylic acid/sulfur) and regular home trimming was recommended. The patient was unable to tolerate oral retinoids (isotretinoin). Initially, physical trimming of footpads under general anesthesia was performed every 4–6 months. However, the frequency by which this was required increased over time, and by the third year of life, it was performed every 8–10 weeks. To address significant discomfort, the patient was also started on gabapentin and codeine for pain control.

Additionally, between 8–12 months of age, the patient developed mild nonseasonal pruritus and recurrent ear infections. The patient was placed on an isoxazoline (Bravecto®) for parasite control. A 10-week strict hydrolyzed protein (Royal Canin Ultamino®) diet trial was performed without improvement, and the patient was presumptively diagnosed with atopic dermatitis. At one year of age, the patient was started on 0.5 mg/kg Oclacitinib (Apoquel®) for control of pruritus.

Despite control of pruritus, the patient continued to develop recurrent ear infections and intermittent episodes of bacterial skin infection. Superficial bacterial infections developed most frequently in areas of heavy wear (elbows, lateral hocks) but were also found occasionally on the trunk, neck, and glabrous areas. At sites of infection, hyperkeratotic plaques with superficial crusting were noted (Appendix A). Infections were managed with topical antiseptics and systemic culture-based antibiotics when indicated.

### 3.2. Histopathological Findings

Histologically, the stratum corneum was markedly expanded by villous projections of orthokeratotic hyperkeratosis. The subtending granular layer of the epidermis was mildly hyperplastic. The samples had no significant inflammation (Figure 2).

### 3.3. Genetic Analysis

We sequenced the genome of the affected dog and searched for homozygous and heterozygous variants in 32 known candidate genes (Table 1) that were not present in the genome sequences of 647 control dogs and 8 wolves (Table 2, Appendix A).

This analysis identified a single homozygous private protein-changing variant in *DSG1*, a known candidate gene for palmoplantar keratoderma in humans [9]. The variant, a 5 bp deletion, can be designated as Chr7:58,163,636_58,163,640del5 (CanFam3.1 assembly). It is a frameshift variant, NM_001002939.1:c.2541_2545delGGGCT, predicted to truncate 207 amino acids from the C-terminus of the wildtype DSG1 protein, NP_001002939.1:p.(Gly848Trpfs*2). We did not investigate whether any mutant protein is expressed or whether the premature stop codon caused by the frameshift deletion leads to nonsense-mediated decay of the transcript.

We confirmed the presence of the 5 bp coding deletion in *DSG1* by Sanger sequencing and genotyped 15 control Rottweiler dogs. The case was homozygous for the mutant allele, while none of the 15 control dogs carried this allele (Figure 3).

## 4. Discussion

In this study, we identified a homozygous *DSG1*:c.2541_2545delGGGCT frameshift variant in a Rottweiler dog with severe footpad hyperkeratosis. *DSG1* encodes desmoglein 1, a calcium-binding transmembrane glycoprotein of the cadherin family. Desmoglein 1 represents a major component of desmosomes that mediates cell–cell adhesion between keratinocytes in the upper layers of the epidermis [45]. Intact desmosomes are essential to maintain the skin barrier function [45]. Desmoglein 1 also represents the major autoantigen in human pemphigus foliaceus [46].

Variants in human *DSG1* cause striate palmoplantar keratoderma I (SPPK1). It is interesting to note that SPKK1 in humans is an autosomal dominant phenotype with the pathogenic variants being present in a heterozygous state in affected individuals [9,47]. In humans, SPKK1 is caused by haploinsufficiency of desmoglein 1 [9,47]. The investigated dog of our study was homozygous for a presumed null allele of *DSG1* and unlikely to express any functional desmoglein 1. In humans, a rare syndromic form of PPK referred to as SAM syndrome has been reported in patients with biallelic *DSG1* loss-of-function variants [48]. Subsequent studies of further human patients with biallelic *DSG1* variants confirmed the dermatitis and multiple allergies but failed to replicate the reported malabsorption and metabolic wasting [49,50]. The clinical presentation of the affected dog in our study, including the development of atopic dermatitis and hyperkeratotic lesions at sites of bacterial skin infection, resembles the phenotype of human SAM syndrome patients without the metabolic wasting, similar to what has been reported in several human cases [49,50].

The histopathological alterations in the footpad skin of the affected Rottweiler were comparable to the changes seen in footpad hyperkeratosis of *FAM83G* mutant Irish Terriers and Kromfohrländer dogs [36] and *KRT16* mutant Dogues de Bordeaux [41]. Thus, the histopathology cannot predict the specific underlying genetic defect. The clinical phenotype of the studied Rottweiler was more severe than in the previously described canine inherited footpad hyperkeratoses [36,41] and required periodical trimming of the excessively hyperkeratotic footpads. In addition to the severely affected footpads, the *DSG1* mutant Rottweiler also had an allergic skin disease and was prone to repeated bacterial skin infections. Such features have not been reported in *FAM83G* mutant Irish Terriers or Kromfohrländer dogs [36] or in *KRT16* mutant Dogues de Bordeaux [41].

Unfortunately, we did not have access to the parents of the affected dog or any other heterozygous dog. It would be interesting to investigate whether heterozygous dogs have completely normal footpads or whether they exhibit a mild phenotype that might go unnoticed by their owners.

In summary, we identified a Rottweiler dog with severe footpad hyperkeratosis that clinically and genetically resembled human SAM syndrome without metabolic wasting. To the best of our knowledge, this dog represents the first nonhuman patient with a spontaneous *DSG1* gene defect.

## Figures and Tables

**Figure 1 genes-11-00469-f001:**
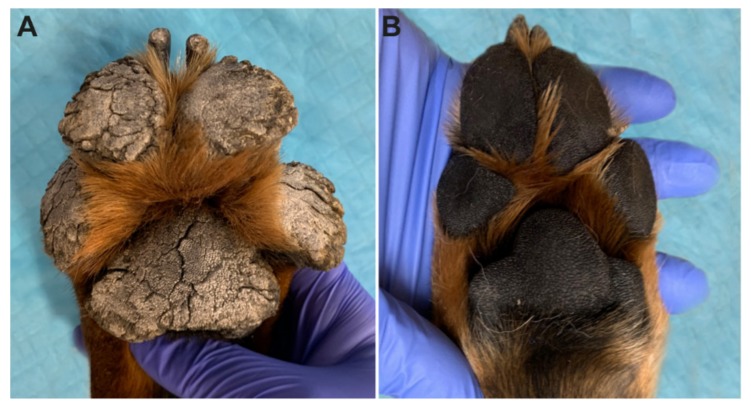
Clinical phenotype at six months of age. (**A**) Marked expansion of the footpads by thick projections of dense keratin in the affected Rottweiler. The adjacent haired skin appears unaffected on this image. (**B**) Footpads of a normal six-month-old control Rottweiler.

**Figure 2 genes-11-00469-f002:**
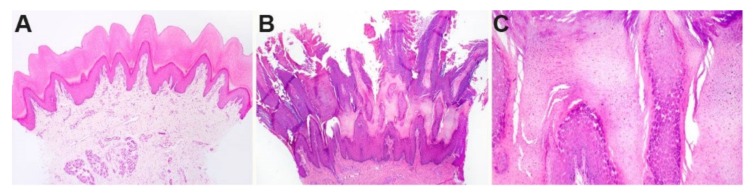
Histopathological phenotype of footpad skin. (**A**) Footpad skin of a normal six-month-old control dog. (**B**) In a biopsy taken from the affected dog at six months of age, a dense proliferation of the stratum corneum (outermost anucleated layer of the skin) markedly expands the epidermis. The stratum corneum is arranged in papillated villous projections of dense keratin. (**C**) Higher magnification of the stratum granulosum and stratum corneum of the affected dog shows an expansion of the granular cell layer.

**Figure 3 genes-11-00469-f003:**
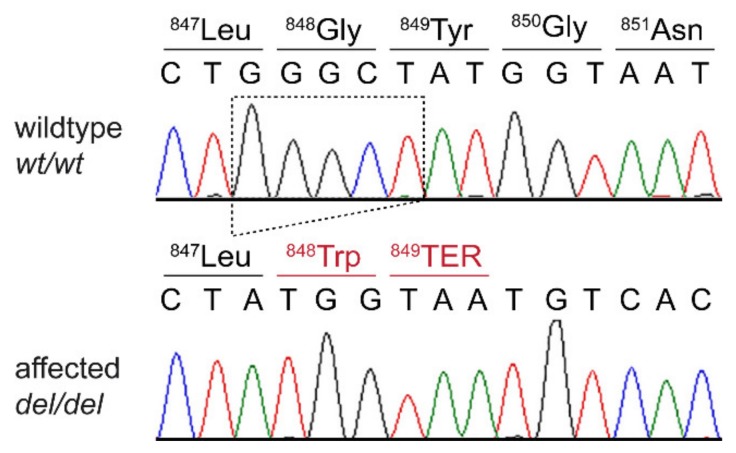
Details of the *DSG1*:c.2541_2545delGGGCT variant. Representative electropherograms of a control and the affected dog are shown. The amino acid translations of the wild-type and mutant alleles are indicated.

**Table 1 genes-11-00469-t001:** Overview of genetic causes of human palmoplantar keratodermas (PPK).

Gene	Phenotype	Inheritance ^a^	Ref.
*AAGAB*	Palmoplantar keratoderma, punctate type IA; PPKP1A	AD	[3,4]
*AQP5*	Palmoplantar keratoderma, Bothnian type	AD	[5]
*CARD5*	Pityriasis rubra pilaris	AD	[6]
*COL14A1*	Palmoplantar keratoderma, punctate type IB; PPKP1B	AD	[7]
*CTSC*	Papillon-Lefevre syndrome	AR	[8]
*DSG1*	Palmoplantar keratoderma I, striate, focal, or diffuse; PPKS1	AD	[9]
*DSP*	Palmoplantar keratoderma II, striate, focal, or diffuse; PPKS2	AD	[10]
*ENPP1*	Cole disease	AD	[11]
*FAM83G*	Palmoplantar keratoderma and exuberant scalp hair.	AR	[12]
*GJA1*	Palmoplantar keratoderma with congenital alopecia	AD	[13]
*GJB2*	Keratoderma, palmoplantar, with deafness	AD	[14]
*GJB3*	Erythrokeratodermia variabilis et progressiva 1	AD or AR	[15]
*GJB4*	Erythrokeratodermia variabilis et progressiva 2	AD	[16]
*GJB6*	Ectodermal dysplasia 2, Clouston type	AD	[17]
*JUP*	Naxos disease	AR	[18]
*KANK2*	Palmoplantar keratoderma and woolly hair	AR	[19]
*KRT1*	Palmoplantar keratoderma, epidermolytic or nonepidermolytic	AD	[20]
*KRT6A*	Pachyonychia congenita 3	AD	[21]
*KRT6B*	Pachyonychia congenita 4	AD	[22]
*KRT6C*	Palmoplantar keratoderma, nonepidermolytic, focal or diffuse	AD	[23]
*KRT9*	Palmoplantar keratoderma, epidermolytic	AD	[24]
*KRT16*	Palmoplantar keratoderma, nonepidermolytic, focal 1, FNEPPK1	AD	[25,26]
*KRT17*	Pachyonychia congenita 2	AD	[26]
*LOR*	Vohwinkel syndrome with ichthyosis	AD	[27]
*POMP*	Keratosis linearis with ichthyosis congenita and sclerosing keratoderma	AR	[28]
*SASH1*	Cancer, alopecia, pigment dyscrasia, onychodystrophy, and keratoderma	AR	[29]
*SERPINB7*	Palmoplantar keratoderma, Nagashima type; PPKN	AR	[30]
*SLURP1*	Meleda disease	AR	[31]
*TAT*	Tyrosinemia, type II	AR	[32]
*TGM1*	Ichthyosis, congenital, autosomal recessive 1	AR	[33]
*TRPV3*	Palmoplantar keratoderma, nonepidermolytic, focal 2; FNEPPK2	AD	[34]
*WNT10A*	Schöpf–Schulz–Passarge syndrome	AR	[35]

^a^ AD: autosomal dominant; AR: autosomal recessive.

**Table 2 genes-11-00469-t002:** Results of variant filtering in the affected Rottweiler dog against 655 control genomes.

Filtering Step	Homozygous Variants	Heterozygous Variants
All variants in the affected Rottweiler	3,310,269	2,516,875
Private variants	842	3290
Protein-changing private variants	4	25
Private variants in known candidate genes	1	0

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
