# Peer review of "A DSG1 Frameshift Variant in a Rottweiler Dog with Footpad Hyperkeratosis"

_genes, 2020, doi:10.3390/genes11040469_

Round 1

Reviewer 1 Report

The authors nicely present a case of presumed hereditary footpad hyperkeratosis in a Rottweiler dog and identify a candidate gene variant. This is a rare but recognized syndrome for which no information is available in the literature for the Rottweiler dog and thus the information provided is novel and valuable. The manuscript is well written and very concise.

Comments

  • The manuscript would be greatly improved by providing molecular evidence of altered DSG1 protein or protein expression in the skin of the patient in comparison to a control patient.
  • Line 126-127: The use of “anucleate keratin” should be replaced with orthokeratotic hyperkeratosis. There is increased stratum corneum, which contains more than keratin.
  • Line 159: DSG1 is the major autoantigen for human pemphigus foliaceus. DSC1 is the major autoantigen in the dog. Please add the word human to this sentence.
  • Please comment on absence or presence of parakeratotic ridges on histopathology as are described for the KRT16 related footpad hyperkeratosis in Dogues de Bordeaux.
  • The manuscript would be improved by the addition of a high magnification image of the stratum granulosum and stratum corneum of the affected dog.

Author Response

(1)

The manuscript would be greatly improved by providing molecular evidence of altered DSG1 protein or protein expression in the skin of the patient in comparison to a control patient.

Response: We agree that molecular evidence at the protein level would further strengthen the manuscript. Unfortunately, as we don’t have an established antibody against the canine DSG1 protein in our laboratories, such an experiment (e.g. immunohistochemistry) will require careful optimization and a considerable amount of time and resources. We are sorry, but this was not possible to achieve in the framework of this study.

(2)

Line 126-127: The use of “anucleate keratin” should be replaced with orthokeratotic hyperkeratosis. There is increased stratum corneum, which contains more than keratin.

Response: Revised accordingly.

(3)

Line 159: DSG1 is the major autoantigen for human pemphigus foliaceus. DSC1 is the major autoantigen in the dog. Please add the word human to this sentence.

Response: Thank you for this important comment. We revised the text accordingly.

(4)

Please comment on absence or presence of parakeratotic ridges on histopathology as are described for the KRT16 related footpad hyperkeratosis in Dogues de Bordeaux.

Response: There are features that resemble both the KRT16 related footpad hyperkeratosis in Dogue de Bordeaux and FAM83G related footpad hyperkeratosis in Irish Terrriers. The granular layer is thickened but proportional to the marked expansion of the stratum corneum. The parakeratosis is a minor feature and should not be confused with the robust parakeratotic hyperkeratosis that is seen in canine metabolic disease (e.g. superficial necrolytic dermatitis, zinc responsive dermatosis). This villous hyperkeratosis also a features of age-related (senile) hyperkeratosis of dogs. It is very important to stress that the histopathology lesions cannot predict the genotype. We added a paragraph comparing the histopathological and clinical findings in the three different canine genodermatoses (DSG1, FAM83G, KRT16) to the discussion.

(5)

The manuscript would be improved by the addition of a high magnification image of the stratum granulosum and stratum corneum of the affected dog.

Response: We added the higher magnification photo and revised figure 2 accordingly.

Reviewer 2 Report

The study is well described, interesting and brings new information to the literature, even if performed on a single subject, therefore practically like a clinical case. The introduction and the materials and methods are well written. In the results, the clinical description is very stringent and it would be better to define it in greater detail, both at the time of the onset of symptoms, and after 8 months, and at 3 years.The discussion is short, but touches on the essential points.

There are some minor revision, listed below

  • keyword: add "dog" and "hyperkeratosis"
  • line 57: add "young" before "male" 
  • line 59: after "causative genetic defect" I think it is necessary to add "for the first time in this dog breed"
  • line 66: It is important to define that all other causes of non-genetic plantar hyperkeratosis were excluded in this subject before inclusion in the study.
  • line 106-109: it'is important to define that no other systemic symptoms or clinical signs were present at the time of the first visit or in the hystory of the subject
  • line 124: no therapy has been set up to manage pruritus or atopic dermatitis?
  • line 125: at what age were biopsies taken? 6 months? it would have been interesting to also biopsy the affected skin at the time of diagnosis of atopy to perform genetic analysis on those samples as well
  • line 154: add that is the first time that a study like this has been done in this breed
  • line 159: add "human" after foliaceus
  • line 168: it is the first time that these hyperkeratotic lesions on the skin have been mentioned. They were not mentioned and described in the clinical examination. add in the results, expanding the clinical description after eight months
  • line 174: add "rottweiler" before dog
  • figure 1 and 2: where were the photos of normal subjects found? do they belong to one of the 15 healthy rottweilers included in the genetic study as a negative control? Why were they biopsied if they had no symptoms? or are they by chance stock photos? if yes, cite the source

Author Response

(1)

keyword: add "dog" and "hyperkeratosis"

Response: To the best of our knowledge, it is not recommended to add words from the title to the keywords. We leave this up to editorial discretion. If title words are allowed in the keywords, then we are certainly happy, if “dog” and “hyerpkeratosis” are added.

(2)

line 57: add "young" before "male"

Response: Revised accordingly.

(3)

line 66: It is important to define that all other causes of non-genetic plantar hyperkeratosis were excluded in this subject before inclusion in the study.

Response: Biopsies were taken at initial presentation to rule out secondary plantar/palmar hyperkeratosis resulting from metabolic, inflammatory or infectious disease. The histopathology features did not reveal any evidence of inflammatory, infectious, caustic or neoplastic processes. We added this information to the methods section (chapter 2.2).

(4)

line 106-109: it is important to define that no other systemic symptoms or clinical signs were present at the time of the first visit or in the hystory of the subject

Response: Revised accordingly.

(5)

line 124: no therapy has been set up to manage pruritus or atopic dermatitis?

Response: Therapy to manage pruritus was established. We expanded chapter 3.1 describing the course and clinical management of the disease accordingly.

(6)

line 125: at what age were biopsies taken? 6 months? it would have been interesting to also biopsy the affected skin at the time of diagnosis of atopy to perform genetic analysis on those samples as well

Response: Yes, the footpad biopsies were taken at 6 months of age. We added this information to the manuscript. We agree that biopsies of the lesional haired skin would be very interesting to study. However, unfortunately, the significance of the atopy and changes of the haired skin was not immediately recognized at the time of clinical diagnosis. We therefore regret that we don’t have access to biopsies of the haired skin of the patient.

(7)

line 154: add that is the first time that a study like this has been done in this breed

Response: We thank the reviewer for this compliment! As we mention at the end of the discussion that this is the first non-human patient with a DSG1 defect (line 178), we felt it inappropriate to over-emphasize the novelty of our study.

(8)

line 159: add "human" after foliaceus

Response: Thank you for this important comment. We revised the text accordingly.

(9)

line 168: it is the first time that these hyperkeratotic lesions on the skin have been mentioned. They were not mentioned and described in the clinical examination. add in the results, expanding the clinical description after eight months.

Response: We expanded the results section accordingly and added a supplementary figure showing such a hyperkeratotic plaque (Figure S1).

(10)

line 174: add "rottweiler" before dog

Response: Revised accordingly.

(11)

figure 1 and 2: where were the photos of normal subjects found? do they belong to one of the 15 healthy rottweilers included in the genetic study as a negative control? Why were they biopsied if they had no symptoms? or are they by chance stock photos? if yes, cite the source

Response: We expanded chapter 2.2. and provide the requested information on the control dogs in Figures 1 & 2.

Reviewer 3 Report

Should clarify in first paragraph of discussion that desmoglein 1 is the major auto-antigen in HUMAN pemphigus foliaceus (not in dogs -- therefore has questionable significance for this report)

With the SAM syndrome in humans being characterized by extensive phenotypic heterogeneity it may be "a stretch" to attribute the pruritus and recurrent ear infections to the DSG1 frameshift in this dog -- figure 1 shows that haired skin is normal in appearance and no mentioned is made of any lesions on the dog other than the footpads.  (

Otherwise very well -written report with information of significance

Author Response

(1)

Should clarify in first paragraph of discussion that desmoglein 1 is the major auto-antigen in HUMAN pemphigus foliaceus (not in dogs -- therefore has questionable significance for this report).

Response: Thank you for this important comment. We revised the text accordingly.

(2)

With the SAM syndrome in humans being characterized by extensive phenotypic heterogeneity it may be "a stretch" to attribute the pruritus and recurrent ear infections to the DSG1 frameshift in this dog -- figure 1 shows that haired skin is normal in appearance and no mentioned is made of any lesions on the dog other than the footpads.

Response: We concede that the description of the clinical phenotype in our original manuscript was somewhat inconsistent. At the initial examination at 6 months of age, the apparent lesions were restricted to the footpads. The phenotypic changes of the haired skin only became apparent starting at 8 months of age and were not properly taken into account when we drafted the original manuscript. We now thoroughly revised the clinical description including the legend to Figure 1 and hope that our manuscript is now more consistent.

Apparently, there is some controversy regarding the human SAM syndrome phenotype and cases with a milder course that much closer resemble our findings in the Rottweiler have been reported. We revised the discussion of the human SAM syndrome phenotype accordingly and included two additional references.